# MiR-34b-5p Mediates the Proliferation and Differentiation of Myoblasts by Targeting IGFBP2

**DOI:** 10.3390/cells8040360

**Published:** 2019-04-17

**Authors:** Zhijun Wang, Xiaocui Zhang, Zhenhui Li, Bahareldin Ali Abdalla, Yangfeng Chen, Qinghua Nie

**Affiliations:** 1Department of Animal Genetics, Breeding and Reproduction, College of Animal Science, South China Agricultural University, Guangzhou 510642, China; zhijunwang@stu.scau.edu.cn (Z.W.); zhangxiaocui@scau.edu.cn (X.Z.); lizhenhui@scau.edu.cn (Z.L.); abdalla406@163.com (B.A.A.); 17620084898@163.com (Y.C.); 2National-Local Joint Engineering Research Center for Livestock Breeding, Guangdong Provincial Key Lab of Agro-Animal Genomics and Molecular Breeding, and Key Laboratory of Chicken Genetics, Breeding and Reproduction, Ministry of Agriculture, Guangzhou 510642, China

**Keywords:** miR-34b-5p, IGFBP2, skeletal muscle, myoblast, myogenesis

## Abstract

As key post-transcriptional regulators, microRNAs (miRNAs) play an indispensable role in skeletal muscle development. Our previous study suggested that miR-34b-5p and IGFBP2 could have a potential role in skeletal muscle growth. Our goal in this study is to explore the function and regulatory mechanism of miR-34b-5p and IGFBP2 in myogenesis. In this study, the dual-luciferase reporter assay and Western blot analysis showed that IGFBP2 is a direct target of miR-34b-5p. Flow cytometric analysis and EdU assay showed that miR-34b-5p could repress the cell cycle progression of myoblasts, and miR-34b-5p could promote the formation of myotubes by promoting the expression of MyHC. On the contrary, the overexpression of IGFBP2 significantly facilitated the proliferation of myoblasts and hampered the formation of myotubes. Together, our results indicate that miR-34b-5p could mediate the proliferation and differentiation of myoblasts by targeting IGFBP2.

## 1. Introduction

MicroRNAs (miRNAs) are small non-coding RNAs which normally can bind to the 3′UTR (3′-Untranslated Region, 3′UTR) of mRNA and function as endogenous translational repressors [1]. Recent studies have proved that miRNAs widely participate in a variety of biological processes such as innate immune responses, cell proliferation and apoptosis, colon, lung, and breast carcinogenesis, and lipid metabolism [2,3,4,5]. In our previous study, we found that the miR-34b-5p was differentially expressed between breast muscle tissues of Recessive White Rock (WRR) and Xinghua Chickens (XH) [6]. Combining this miRNA-Seq data with our previous RNA-Seq data [7], we predicted that the differentially expressed gene *IGFBP2* could be the target gene of miR-34b-5p. Thus, we speculated that miR-34b-5p and IGFBP2 could play a potential role in broiler growth.

MiR-34b-5p is a member of the miR-34 family (miR-34a, miR-34b and miR-34c) and can be processed to its precursor miRNA (miR-34b) with a mature sequence of 23 nucleotides. The miR-34 family is a well-known cancer suppressor, which can be induced by p53 and participates in many cancer biological processes [8,9,10]. In chicken, it has been reported that miR-34b-5p can promote the proliferation of ALV-J-infected cells and ALV-J replication by targeting MDA5 [11]. Besides cancer, miR-34 was differentially expressed miRNA during the development of skeletal muscle in Nile tilapia [12]. All members of the miR-34 family could be involved in and affect osteoblast differentiation during embryonic development and bone-mass accrual after birth in mice [13,14]. miR-34a can also promote the proliferation of human pulmonary artery smooth muscle cells [15]. In addition, miR-34a and miR-34c could inhibit the proliferation of smooth muscle cells [16,17]. miR-34c was also reported to repress the proliferation of muscle satellite cells while promoting the differentiation process [18]. All these studies indicate that miR-34 not only can mediate the physiological process of cancer, but also can participate in the regulation of muscle growth.

Insulin-like growth factor binding protein 2 (IGFBP2) is a member of the IGFBPs (IGFBP1-6) family. These six homologous proteins are famous for their high-affinity binding to IGF1 and IGF2 [19]. IGFBP2 can participate in many different kinds of cancer as a potential biomarker [20]. Previous research has found that IGFBP2 was abundantly expressed in the eyes, brain, skeletal muscle, and heart of chickens during the embryonic period [21], and highly expressed in embryonic tissues in mammals [22]. The DNA polymorphism of IGFBP2 was significantly correlated with growth, carcass, meat quality, and reproductive traits of pigs [23,24]. However, in addition to the above studies, few reports have been made about IGFBP2 in poultry skeletal muscle development.

In this study, we explored the functional role and regulation of miR-34b-5p and IGFBP2 in chicken skeletal muscle development. We confirmed that miR-34b-5p could suppress the cell cycle progression of myoblast and accelerate the formation of myotubes by directly repressing the expression of IGFBP2.

## 2. Materials and Methods

### 2.1. Animals and Cells

The tissues including heart, liver, spleen, lung, kidney, breast muscle, and leg muscle were collected from female Xinghua chickens at 7 weeks of age. The chicken primary myoblasts (CPM) were isolated from leg muscle tissues of Yuhe chickens in at an embryo age of 11 (Zhuhai Yuhe Company Ltd, Zhuhai, China), following the method that we previously reported [25]. All experimental animals were operated on in accordance with approved guidelines of the Animal Care and Use Committee of the South China Agricultural University (Approval number: SCAU#0014). Please provide the ethic code.

### 2.2. Cell Culture and Transfection

DF-1 cell lines were cultured in Dulbecco’s modified Eagle’s medium (Gibco, Grand Island, NY, USA) supplemented with 10% fetal bovine serum (FBS) (Gibco, Grand Island, NY, USA) and 0.5% penicillin/streptomycin (Invitrogen, Carlsbad, CA, USA). CPM cells were cultured in growth medium (GM, Roswell Park Memorial Institute (RPMI)-1640 medium with 20% FBS and 0.5% penicillin/streptomycin) and induced to differentiate in differentiation medium (DM, FBS reduced to 5%).

All the RNA oligonucleotides used in this study, including miR-34b-5p mimics, miR-34b-5p inhibitor, and si-gga-IGFBP2 (small interfering RNA (siRNA) used for the knockdown of IGFBP2), were obtained from RiboBio (Guangzhou, China) and the sequences are showed in Table 1. The transfection concentrations of these three oligonucleotides were 10 nM, 100 nM, and 100 nM. The oligonucleotides were transfected using Lipofectamine 3000 reagent (Invitrogen, USA) according to the manufacturer’s protocol with at least three replications.

### 2.3. Plasmid Construction

Primers used in this study were designed using Premier Primer 5.0 software (Premier Biosoft International, Palo Alto, CA, USA), and synthesized by Sangon Biotech (Shanghai, China).

PmirGLO dual-luciferase reporter vector: The 3′UTR fragment of *IGFBP2* (NCBI (National Center for Biotechnology Information) Reference Sequence: NM_205359.1) containing the miR-34b-5p binding site was cloned into the pmirGLO vector through the PMD^TM^-18T cloning vector, along with the mutation vector (Takara, Ostu, Japan). The primers were named IGFBP2-WT and IGFBP2-MT and the sequences are listed in Table 2.

The IGFBP2 overexpression vector: The full coding sequence (CDS) of IGFBP2 was synthesized by Sangon Biotech (Shanghai, China) and cloned into the pcDNA3.1 expression vector.

### 2.4. Dual-Luciferase Reporter Assay

The pmirGLO dual-luciferase reporter vector (200 ng) containing a wild-type or mutant IGFBP2-3′UTR fragment was co-transfected with miR-34b-5p mimic or NC (Negative Control) duplexes (100 nM) into DF-1 cells in a 96-well plate with six independent repeats. After 48 h of transfection, Firefly and Renilla luciferase activities were measured in a Fluorescence/Multi-Detection Microplate Reader (BioTek, Winooski, VT, USA) using a Dual-GLO Luciferase Assay System Kit (Promega, Madison, WI, USA).

### 2.5. RNA Isolation, Complementary DNA (cDNA) Synthesis, and Quantitative Real-Time PCR (qRT-PCR)

Total RNA was isolated from skeletal muscle tissues or cells using RNAiso Plus (Takara, Ostu, Japan). cDNA synthesis for mRNA was carried out using a PrimeScript RT Reagent Kit with gDNA Eraser (Perfect Real Time) (Takara, Japan). The reverse transcription reaction for miRNA was performed with ReverTra Ace qPCR RT Kit (Toyobo, Osaka, Japan) using specific Bulge-loop miRNA qRT-PCR Primer for miR-34b-5p and U6 designed by RiboBio (Guangzhou, China). qRT-PCR reactions were carried out in a QuantStudio 5 Real-Time PCR Systems (Thermo Fisher, Waltham, MA, USA) with iTaq Universal SYBR Green Supermix Kit (Toyobo, Japan). The comparative 2^−ΔΔ*C*T^ method [26] was used for qRT-PCR data analysis. The primers used for qRT-PCR are also listed in Table 2.

### 2.6. EdU (5-Ethynyl-2′-Deoxyuridine) Assay

After 2 h of incubation with 50 μM 5-ethynyl-2′-deoxyuridine (EdU; RiboBio, China), the CPM cells were fixed and stained with a C10310 EdU Apollo In Vitro Imaging Kit (RiboBio, China). We used a Leica DMi8 fluorescent microscope to capture the EdU-stained cells with three randomly selected fields. The proliferation rate was calculated by the ratio of the number of EdU-stained cells to the number of Hoechst 33342-stained cells.

### 2.7. Flow Cytometric Analysis

After 48 h of transfection, the CPM cells cultured in 12-well plates were collected and fixed in 70% ethanol at −20 °C overnight. The flow cytometric analysis was carried on a BD AccuriC6 flow cytometer (BD Biosciences, San Jose, CA, USA) with a Cell Cycle Analysis Kit (Thermo Fisher Scientific, Waltham, MA, USA). The data were processed using the FlowJo7.6 software (Treestar Incorporated, Ashland, OR, USA).

### 2.8. Western Blotting

Radio-immunoprecipitation assay (RIPA) buffer and phenylmenthanesulfonyl fluoride (PMSF) protease inhibitor were used here to extract the CPM cellular proteins. The Western blot assays were carried out as previously reported [27]. The primary antibodies used in this study were as follows: IGFBP2 rabbit polyclonal antibody (11065-3-AP; proteintech, Chicago, IL, USA; 1:300), MyHC mouse monoclonal antibody (B103; DHSB, Lowa City, IA, USA; 1:500), and β-Tubulin monoclonal antibody (A01030; Abbkine, California, CA, USA; 1:5000). HRP conjugated goat anti-rabbit IgG (A21020; Abbkine, USA; 1:10000) and HRP conjugated goat anti-mouse IgG (A21010; Abbkine, USA; 1:10000) were used as secondary antibodies.

### 2.9. Immunofluorescence

After 48 h of transfection, CPM cells cultured in 12-well plates were first treated with 4% formaldehyde for 20 min and then permeabilized by 0.1% Triton X-100. Next, after being blocked for 30 min with goat serum, the cells were then incubated overnight with anti-MyHC (B103; DHSB, USA; 1:50). The cells were treated with ProteinFind goat anti-mouse IgG (H+L) FITC conjugate (HS221; Transgen, China; 1:200) or goat anti-mouse IgG (H+L)-Dylight 594 (BS10027; Bioworld, USA; 1:200) for 1 h. The cell nuclei were stained with DAPI (Beyotime, Shanghai, China) for 5 min and then images were captured with a Leica DMi8 fluorescent microscope (Leica, Wetzlar, Germary). We calculated the percentage of the total image area covered by myotubes as the total myotube area by using the ImageJ software (National Institutes of Health, Bethesda, MD, USA).

### 2.10. Statistical Analysis

All experimental results are presented as the mean ± S.E.M, with at least three independent replications. The statistically significant difference between groups was tested by independent sample t-test. We considered *P* < 0.05 to be statistically significant. * *P* < 0.05; ** *P* < 0.01; *** *P* < 0.001.

## 3. Results

### 3.1. miR-34b-5p Represses the Cell Cycle Progression of Myoblasts

In order to reveal the function of miR-34b-5p in myoblasts, we transfected the miR-34b-5p mimic and inhibitor into CPM cells to assess its effect on cell cycle progression (Figure 1A). The EdU assay found that the overexpression of miR-34b-5p could hamper the proliferation of CPM cells and the opposite results could be found when we inhibited the expression of miR-34b-5p (Figure 1B,C). In addition, the cells in G0/G1 phase were significantly increased after being transfected with miR-34b-5p mimic, while the number of cells in S phase were lower than that in the control group (Figure 1D,F). The cell cycle changes transfected with miR-34b-5p inhibitor showed the opposite trend (Figure 1E,F). Collectively, these results demonstrate that miR-34b-5p could repress the cell cycle progression of myoblasts.

### 3.2. miR-34b-5p Promotes the Formation of Myotubes

miR-34b-5p showed a relatively high expression in kidney, breast muscle, and leg muscle of 7 week old Xinghua chickens (Figure 2A). CPM cells at around 90% confluency were induced to undergo differentiation by changing the growth media (GM) to differentiation media (DM), as previously described. GM represents myoblasts in the proliferative phase, while DM1–DM6 represent the myoblasts that were successfully induced to differentiate from day 1 to day 6 (at different time points). Results showed that miR-34b-5p expression increased abruptly at middle (DM3) and late (DM5) differentiation stages (Figure 2B). Therefore, we transfected miR-34b-5p mimic into CPM cells and found that the overexpression of miR-34b-5p remarkably increased the expression of MyoD (myogenic differentiation), MyoG (myogenin), and MyHC (myosin heavy chain) (three important marker genes during myogenesis [28,29,30,31,32]), whereas the inhibition of miR-34b-5p had an opposite effect on these genes (Figure 2E,F). The Western blot analysis showed that miR-34b-5p could increase the expression of MyHC (Figure 2G,H). Also, the MyHC staining results showed that the number and the area of myotubes were greatly increased after miR-34b-5p mimic transfection, while the knockdown of miR-34b-5p could inhibit the formation of myotubes (Figure 2C,D). These results suggest that miR-34b-5p promotes myoblast differentiation.

### 3.3. IGFBP2 is a Direct Target of miR-34b-5p

The miR-34b-5p binding site in the 3′UTR region of *IGFBP2* was successfully mutated (Figure 3A,B). A dual-luciferase reporter assay carried out in DF-1 cells showed that miR-34b-5p could reduce the luciferase activity of pmirGLO-IGFBP2-WT compared with the NC group (Figure 3E). The mRNA and protein level of IGFBP2 showed that miR-34b-5p could directly inhibit the expression of IGFBP2 (Figure 3C,D).

### 3.4. IGFBP2 Facilitates the Cell Cycle Progression of Myoblasts

IGFBP2 had a relatively high expression in kidney, breast muscle, and leg muscle of 7 week old Xinghua chickens (Figure 4A), similar to the miR-34b-5p expression (see Figure 2A). The overexpression plasmid (pcDNA3.1-IGFBP2) and si-IGFBP2 efficiency were detected by qRT-PCR and Western blot analysis. These results confirm that IGFBP2 was successfully overexpressed and knocked-down in CPM cells (Figure 4B,C). The EdU assay showed that the number of cells in the proliferative phase was significantly increased due to the overexpression of IGFBP2, and decreased after *IGFBP2* knockdown (Figure 4D,E). The number of cells in G0/G1 phase were less than that in the control group after IGFBP2 overexpression, and the contrary trend could be found after *IGFBP2* knockdown (Figure 4F–H).

### 3.5. IGFBP2 is an Inhibitor of Myotube Formation

qRT-PCR results indicate that IGFBP2 could have a potential role during myoblast differentiation, since it showed a relatively high expression level during myoblast differentiation, especially at DM2 and DM4 (Figure 5A). Meanwhile, the mRNA expression of MyoD and MyHC was downregulated by IGFBP2 overexpression (Figure 5B), and upregulated after IGFBP2 inhibition (Figure 5C). The protein level of MyHC was inhibited by IGFBP2 and upregulated after IGFBP2 knockdown (Figure 5F,G). Immunofluorescence staining showed that the formation of myotubes could be hampered by IGFBP2 overexpression; conversely, IGFBP2 knockdown could promote the formation of myotubes (Figure 5D,E). These results suggest that IGFBP2 could serve as an inhibitor of myotube formation.

## 4. Discussion

Myoblast proliferation, differentiation, and fusion into multinucleated myotubes occur to form myofibers, and the formation of myofibers is one of the processes of skeletal muscle development that only happens in the embryonic stage [28,29]. Besides some muscle-specific miRNAs (such as miR-206, miR-1, and miR-133), many non-muscle-specific miRNAs are also required for the differentiation of myoblasts. This is the reason why we chose chicken primary myoblasts to investigate the function of miR-34b-5p and IGFBP2.

It has been reported that miR-34 can target many genes involved in cell cycle progression, such as *CCND1*, *CCNE2*, *CDK4*, *CDK6*, *E2F3*, *MAP2K1*, *MAP3K9*, *MYC*, and so on [30], and our present study with gga-miR-34b-5p shared the same binding sites with hsa-miR-34b-5p. This means that gga-miR-34b-5p could possibly mediate the cell cycle progression through these genes. In our study, we found that miR-34b-5p overexpression increased the proportion of cells in G0/G1 phase and reduced the proportion of cells in S phase, suggesting that it can suppress cells from G0/G1 to S phase in CPM. It was reported that miR-34b could repress muscle cell proliferation and promote myotube formation by targeting NUCL in mouse myoblast C2C12 cells [31], which proved that our results are true and reliable. Previous studies have found that miR-34b-5p could induce G1/S shift by targeting MDA5 in DF-1 cells, while in our study, we found that IGFBP2 could facilitate the cell cycle progression of myoblasts. This difference may be explained by a study that showed that miR-34b-5p had a totally different effect when compared between DF-1 and myoblast cells [11]. Our results in Figure 2A indicate that miR-34b-5p could also play a potential regulatory role in postnatal myofiber hypertrophy. MyoD and MyoG are important marker genes during myogenesis [32,33,34]. The major difference between muscle fiber types is related to their myosin complement [35]. So, myosin heavy chain (MyHC) staining would be an appropriate way to describe myoblast differentiation. Here, we confirmed that miR-34b-5p could accelerate the formation of myotubes by promoting the mRNA level of MyoD, MyoG, and MyHC as well as the protein level of MyHC.

The role of IGFBP2 has been studied in angiogenesis [36] and many types of cancers [20], but not in chicken skeletal muscle development. In the present study, we proved that IGFBP2 had a relatively high expression in the breast muscle and leg muscle of 7 week old Xinghua chickens (Figure 4A) and it could participate in the proliferation and differentiation of myoblasts. We used chicken primary myoblast cells to better understand the how IGFBP2 regulates skeletal muscle development. Previous studies have found that the DNA polymorphism of IGFBP2 is associated with chicken growth, body composition, and carcass traits [37,38]. In this study, we found that IGFBP2 could facilitate the cell cycle progression of myoblasts by promoting the transition of myoblasts from G0/G1 phase to S phase, and could suppress the formation of myotubes by inhibiting MyoD and MyHC. It is also reported that c-Myc and ZEB1 could be repressors of myoblast differentiation [39,40], but we found that IGFBP2 could repress the expression of these two genes (data not shown), indicating that IGFBP2 might play a more complex regulatory role in the differentiation of myoblasts than we expected.

## 5. Conclusions

In conclusion, we found that miR-34b-5p could repress the proliferation and promote the differentiation of myoblasts by targeting IGFBP2 (Figure 6).

## Figures and Tables

**Figure 1 cells-08-00360-f001:**
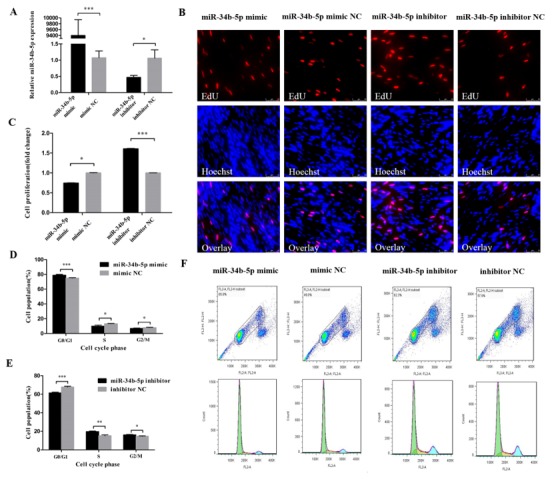
miR-34b-5p represses the cell cycle progression of myoblasts. (**A**) The transfection efficiency of miR-34b-5p after overexpression and inhibition of miR-34b-5p. (**B**) EdU staining after the transfection of miR-34b-5p mimic and inhibitor in chicken primary myoblasts (CPM) cells. (**C**) The proliferation rate of CPM cells transfected with miR-34b-5p mimic and inhibitor. (**D**,**E**) The statistical results of cell cycle analysis after overexpression and inhibition of miR-34b-5p in myoblast. (**F**) Flow cytometry raw data of cell cycle analysis in myoblasts after overexpression or inhibition of miR-34b-5p. Results are shown as mean ± S.E.M. and the data are representative of at least three independent assays. Independent sample t-test was used to analyze the statistical differences between groups. (* *P* < 0.05; ** *P* < 0.01, *** *P* < 0.001). Please provide a sharper figure.

**Figure 2 cells-08-00360-f002:**
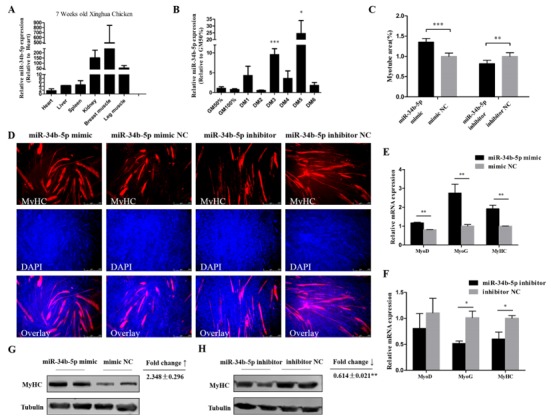
miR-34b-5p promotes the formation of myotubes. (**A**) The relative expression of miR-34b-5p in six tissues of 7 week old Xinghua chickens. (**B**) The relative expression of miR-34b-5p during CPM differentiation. GM represents myoblasts in the proliferative phase, while DM1–DM6 represent myoblasts that were successfully induced to differentiate from day 1 to day 6. (**C**) Myotube area (%) of CPM cells 72 h after overexpression and inhibition of miR-34b-5p. (**D**) MyHC staining of CPM cells 72 h after transfection of miR-34b-5p mimic and inhibitor. (**E**,**F**) MicroRNA (mRNA) level of *MyoD*, *MyoG*, and *MyHC* after transfection with miR-34b-5p mimic or inhibitor in CPM cells. (**G**,**H**) The protein expression of MyHC was determined by Western blotting in CPM cells after transfection with miR-34b-5p mimic and inhibitor. Results are shown as mean ± S.E.M. and the data are representative of at least three independent assays. Independent sample t-test was used to analyze the statistical differences between groups. (* *P* < 0.05; ** *P* < 0.01, *** *P* < 0.001).

**Figure 3 cells-08-00360-f003:**
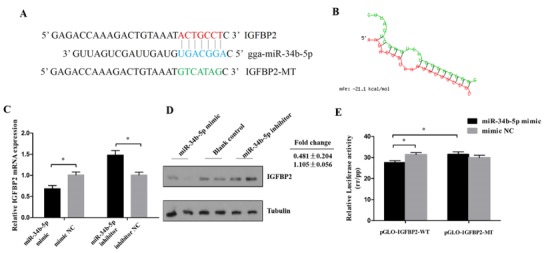
IGFBP2 is a direct target of miR-34b-5p. (**A**) The miR-34b-5p binding site in the *IGFBP2* mRNA 3′UTR. The seed sequence and mutant sequence in miR-34b-5p were highlighted in blue and green, respectively. (**B**) The potential miR-34b-5p targeting site in IGFBP2 mRNA 3′UTR was analyzed by RNAhybrid software. (**C**,**D**) The mRNA and protein level of IGFBP2 in CPM cells after being transfected with miR-34b-5p mimic or inhibitor. (**E**) The dual-luciferase reporter assay was performed by co-transfecting a wild-type or mutant *IGFBP2* 3′UTR with miR-34b-5p mimic or mimic NC in DF-1 cells. Results are shown as mean ± S.E.M. and the data are representative of at least three independent assays. Independent sample t-test was used to analyze the statistical differences between groups. (* *P* < 0.05).

**Figure 4 cells-08-00360-f004:**
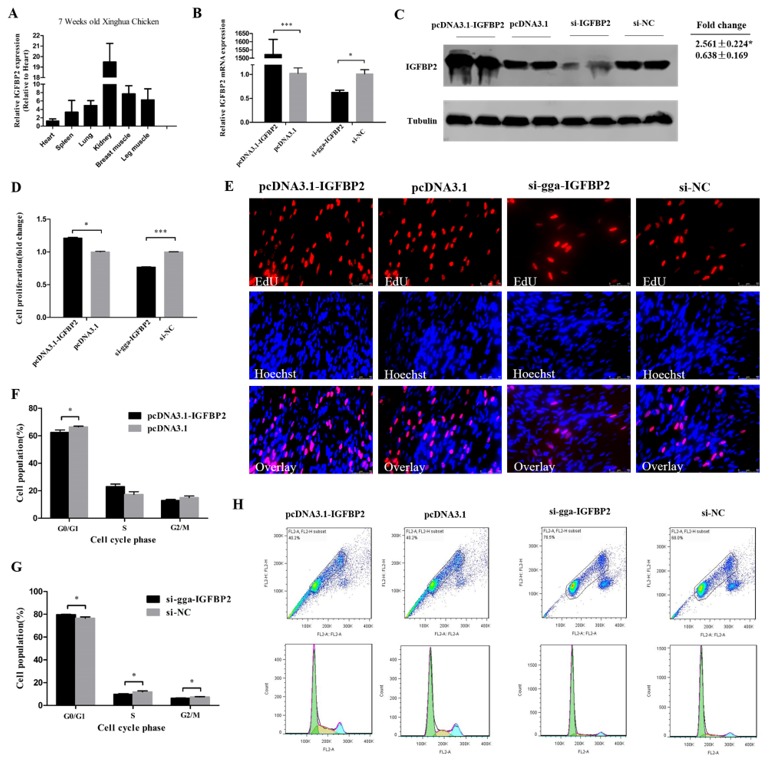
IGFBP2 facilitates the cell cycle progression of myoblasts. (**A**) The relative mRNA expression of IGFBP2 in six different tissues of 7 week old Xinghua chickens. (**B**,**C**) The mRNA and protein level of IGFBP2 after IGFBP2 overexpression and knockdown. (**D**,**E**) The proliferation rate and EdU assay of CPM cells after overexpression and inhibition of IGFBP2. (**F**–**H**) Flow cytometry raw data and statistical results of cell cycle analysis after IGFBP2 overexpression or knockdown in myoblasts. Results are shown as mean ± S.E.M. and the data are representative of at least three independent assays. Independent sample t-test was used to analyze the statistical differences between groups. (* *P* < 0.05; *** *P* < 0.001).

**Figure 5 cells-08-00360-f005:**
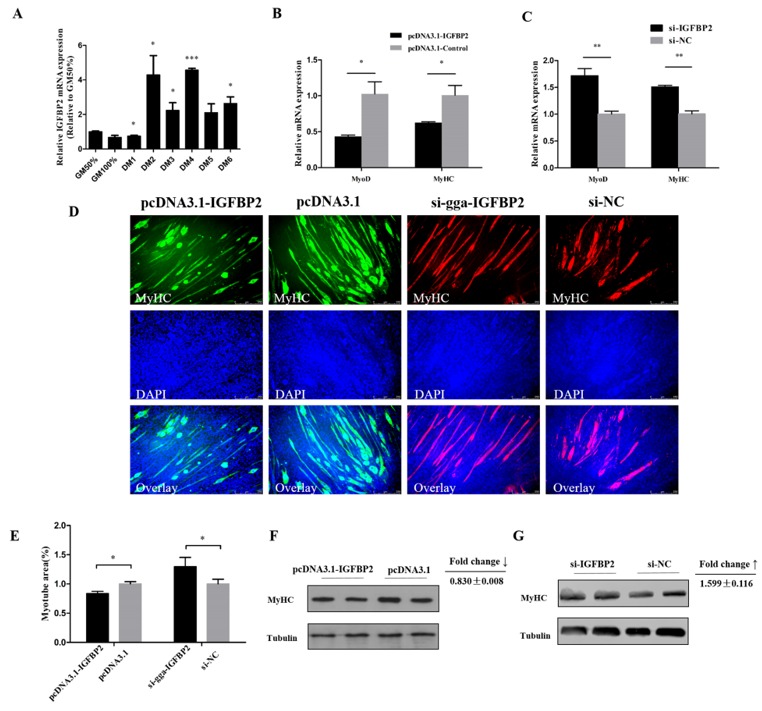
IGFBP2 suppresses myotube formation. (**A**) The relative expression of IGFBP2 during CPM differentiation. (**B**,**C**) The mRNA level of MyoD and MyHC after overexpression or knockdown of IGFBP2 in CPM cells. (**D**,**E**) MyHC staining and the myotube area (%) of CPM cells 72 h after overexpression and knockdown of IGFBP2. (**F**,**G**) The protein level of MyHC after overexpression and inhibition of IGFBP2. Results are shown as mean ± S.E.M. and the data are representative of at least three independent assays. Independent sample t-test was used to analyze the statistical differences between groups. (* *P* < 0.05; ** *P* < 0.01, *** *P* < 0.001).

**Figure 6 cells-08-00360-f006:**
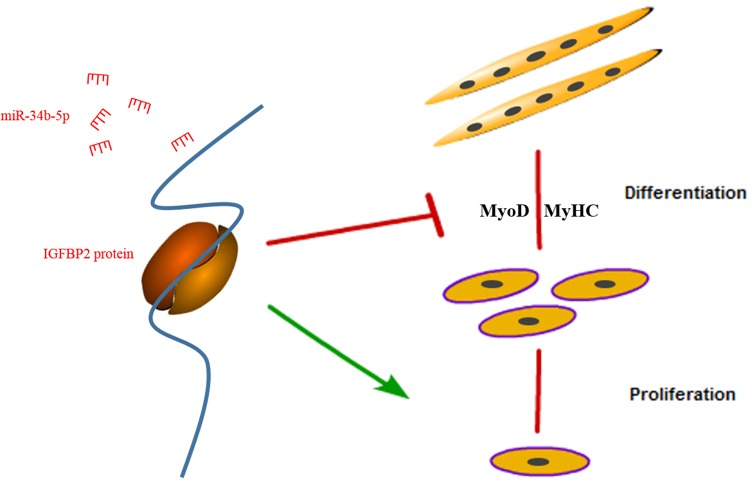
A schematic model of an IGFBP2-mediated regulatory network during myoblast proliferation and differentiation.

**Table 1 cells-08-00360-t001:** Oligonucleotide sequences used in this study.

Fragment Name	Sequence (5′ to 3′)
miR-34b-5p mimic	CAGGCAGUGUAGUUAGCUGAUUG
miR-34b-5p inhibitor	CUUUCUGCUUUCUUCUCUGCCUG
si-gga-IGFBP2	GGGAGTGTCTCTCTTTCTT

**Table 2 cells-08-00360-t002:** Primers used in this study.

Primer Name	Primer Sequences (5′ to 3′)	Size (bp)
*IGFBP2-WT*	F:ACCATTTCCCTCTTCCTCC	295
	R:ACCAAGCATTCAGCTCCAC	
*IGFBP2-MT*	F:CGAGACCAAAGACTGTAAATTGTGAGTCTTGTGTCCTGCC	2987
	R:GGCAGGACACAAGACTCACAATTTACAGTCTTTGGTCTCG	
*IGFBP2*	F:AGCGGCAGATGGGCAAAGT	184
	R:GGGGATGTGGAGGGAGTAGAGG	
*β-actin*	F:TCATTGTGCTAGGTGCCA	160
	R:TCATTGTGCTAGGTGCCA	
*MyoD*	F: GCTACTACACGGAATCACCAAAT	200
	R: CTGGGCTCCACTGTCACTCA	
*MyoG*	F: CGGAGGCTGAAGAAGGTGAA	320
	R: CGGTCCTCTGCCTGGTCAT	
*MyHC*	F: CTCCTCACGCTTTGGTAA	213
	R: TGATAGTCGTATGGGTTGGT	

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
