# Peer review of "MiR-34b-5p Mediates the Proliferation and Differentiation of Myoblasts by Targeting IGFBP2"

_cells, 2019, doi:10.3390/cells8040360_

Round 1
Reviewer 1 Report
Fig 1, 2, 4, 5. The quality of the fluorescence (especially of the Hoechst) is to be improved
Fig 2. Western blot G and H need quantitative evaluation
Fig 4. Western blot C requires quantitative evaluation and should be replaced if possible.
Fig 5. Western blot F and G require quantitative evaluation
Author Response
Point 1: Fig 1, 2, 4, 5. The quality of the fluorescence (especially of the Hoechst) is to be improved.
Response 1: Thank you for your suggestion, we have improved the quality of all fluorescence images, and we hope that this version will meet your requirements.
Point 2: Fig 2. Western blot G and H need quantitative evaluation.
Fig 4. Western blot C requires quantitative evaluation and should be replaced if possible.
Fig 5. Western blot F and G require quantitative evaluation
Response 2: Thank you for your suggestion, all of our western blot results have supplemented the quantitative evaluation including Fig 3D. And also, Fig 4C have been replaced by a new western blot results.
Reviewer 2 Report
The manuscript entitled "MiR-34b-5p Mediates The Proliferation And 3 Differentiation Of Myoblast By Targeting IGFBP2" by Wang and colleagues, aimed to explore the function and regulatory mechanisms of miR-34b-5p and IGFBP2 in myogenesis. Although the data is interesting and relevant to the field, I do have some concerns regard the study.
While the data is intriguing, the amount of mimic used for transfection is too high in my opinion. It is known that when you have a supraphysiological dose of mimic and/or inhibitor, you can alter nonspecific targets and compromise the whole system. The authors should provide a dose-response curve to determine if 100nM is actually appropriate. A 9400 fold increase could be too much for the cells, causing changes in a nonspecific way. The authors should, at least, provide some explanation of what this dose was used.
Author Response
Point 1: While the data is intriguing, the amount of mimic used for transfection is too high in my opinion. It is known that when you have a supraphysiological dose of mimic and/or inhibitor, you can alter nonspecific targets and compromise the whole system. The authors should provide a dose-response curve to determine if 100nM is actually appropriate. A 9400 fold increase could be too much for the cells, causing changes in a nonspecific way. The authors should, at least, provide some explanation of what this dose was used.
Response 1: Thank you for your comments. The transfection concentration of miR-34b-5p mimic we used in this study was 10nM, and the transfection concentration of miR-34b-5p inhibitor and si-gga-IGFBP2 were 100nM. In our pre experiment, we transfected miR-34b-5p in a concentration gradient in 10nM, 25nM, 50nM, and 100nM respectively, and our qRT-PCR results showed that they all can have an almost 9000 fold increase and could inhibit the expression of IGFBP2. So we chose the minimum transfection concentration as 10nM. In Fig 2B, we found that miR-34b-5p expression increases abruptly at middle (DM3) and late (DM5) differentiation stage, while in GM100% the expression is particularly low, and the overexpression of miR-34b-5p in myoblast cells were collected almost at GM100%, so we conjectured that it may be one of the reason caused a such high fold increase.
According to your suggestion, we have revised the manuscript extensively, and we hope that this version will meet your requirements.
Round 2
Reviewer 2 Report
The authors have addressed all of my concerns.
This manuscript is a resubmission of an earlier submission. The following is a list of the peer review reports and author responses from that submission.
Round 1
Reviewer 1 Report
The authors report that MiR-34b-5p targets IGFBP2 and represses differentiation of myoblasts. The findings are potentially interesting, but the reviewers have two major concerns.
1. The numbers and characters in the figures are too small to read. Therefore, it is impossible to evaluate this manuscript.
2. Second, in Fig. 2B, the expression pattern of miR-34b-5p is strange. There is no explanation on this periodic pattern. The authors should confirm this by repeating experiments.
Minor points.
1. It would be more informative to show fusion index in addition to the myotube area.
2. Fig.3F. Is there any significant difference in luciferase activity between pGLO-IGFBP2-WT and pGLO-IGFBP2-MT?
3. Line 242. The reviewer cannot understand why the data shown in Fig.2A indicate that MiR-34b-5p is related to muscle hypertrophy.
Author Response
Major points.
1. We are very sorry for our carelessness. In the revised version, all figures have been revised and improved, and we hope they will meet your requirements.
2. We know that miRNAs are small and unstable, and the expression of miRNAs is very sensitive and has no fixed pattern, for example, the expression of miR-17-5p, miR-20a, miR-93 and miR-106a in differentiating embryonic stem cells could increase or decrease at different time [1]. In this study, miR-34b-5p showed a relatively high expression in DM3 and DM5 but low in DM4 and DM6, probably because that miR-34b-5p not only could play an important role in myoblast differentiation, but also could be regulated by other regulatory factors to ensure that the differentiation process can proceed normally. As for repetitive experiments, we could not perform additional experiments in only 10 days given by editor for our revision, and moreover we believe that our data are accurate and reliable.
Ref:
[1] Foshay KM.Gallicano GI. miR-17 family miRNAs are expressed during early mammalian development and regulate stem cell differentiation. DEV BIOL. 2009, 326,431-443. doi:10.1016/j.ydbio.2008.11.016
Minor points.
1. In order to show the integrity of the myotube, the magnification of the microscope is only 100X, also the cell density is too high to count, so we could not calculate the fusion index of the myotubes in our immunofluorescence.
2. Yes, the luciferase activity of pGLO-IGFBP2-WT is significant different with pGLO-IGFBP2-MT, and this figure has been revised and improved as you suggested in Fig.3E.
3. The formation of myofiber mainly occurs in embryogenesis, and the hypertrophy may undergo after the stage that myofibers are formed at the postnatal stage [2]. So in Fig.2A we found that miR-34b-5p showed a relative high expression at 7 weeks Xinghua chicken in breast and leg muscle, we conjectured that it may relate to muscle hypertrophy, but there might be other possible regulatory effects.
Ref:
[2] LUO W.; ABDALLA BA.; NIE Q.ZHANG X. The genetic regulation of skeletal muscle development: insights from chicken studies. Frontiers of Agricultural Science and Engineering. 2017, 4,295. doi:10.15302/J-FASE-2017159
Reviewer 2 Report
Major problems
MiR-34b-5p it is known to be pro proliferative you must explain, the same in terms of cell cycle in DF-1 cell induces G1/S shift (ref. 11)
The role of MiR-34b-5p as promoter of myoblast differentiation is in contrast with the proliferative role, see above
You said "IGFBP2 had a relatively high expression in kidney, breast muscle and leg muscle of 7 weeks 195 Xinghua Chicken (Figure 4A), just like miR-34b-5p" ;But if IGFBP2 is target of MiR-34b-5p the expression should not be in contrast?
The history of IGFBP2 and the differentiation is based on contrasting observations .... in some cases it is pro in others it is against
SHARPLES et al. 2010
Minor
Fig1.Control that demonstrates over-expression and inhibition of the MiR-34b-5p is missing
Image quality too low, how did you count nuclei and positive EdU cells?
Fig2.When you introduce a marker or an indication related to an experiment you have to explain what it is (eg DM3 and DM5, MyoD, MyoG and MyHC)
Image quality too low, Magnification?
Fig3. What happens to the proliferation of DF-1? Show me the antiproliferative effects when over express MiR-34b-5p
Fig4 (B, D) Show me the over expression of IGFBP2
Image quality too low
Fig5 Image quality too low, Magnification?
Fig6 It seems that mrna is responsible for the effects not that it is the decrease of the protein
Author Response
Major points.
1. Many genes and transcription factors involved in cell cycle and differentiation progress could be the target genes of miR-34 family, therefore the functions of miR-34b-5p in cell proliferation and differentiation mainly depend on different target genes. In DF-1 cell lines, miR-34b-5p could induce G1/S shift by targeting MDA5, while in our study, we found that IGFBP2 could facilitate the cell cycle progress of myoblast and inhibit myotube formation, which could probably explain that miR-34b-5p showed a totally different function between DF-1 and myoblast cells. This explanation has also been added into discussion part in line 248-251.
2. In Fig.2A and Fig.4A we found a relative expression of miR-34b-5p and IGFBP2, which just means they are relative abundant in these three tissues in comparison with other tissues. If we want to know their true expression level, maybe we should try absolute quantification.
3. We believed that one gene can have multiple effects, so does IGFBP2. In line 267-270 of our discussion part, we have mentioned that IGFBP2 could repress the expression of c-Myc and ZEB1, while these two genes happened to be the repressor of myoblast differentiation, which means sometimes IGFBP2 can also promote the differentiation of myoblast. The process of skeletal muscle development is complex and precise, and more evidence is required to prove that IGFBP2 could regulate skeletal muscle development through different pathways and genes. And in this study, we found that IGFBP2 could repress the formation of myotube by directly inhibiting the expression of MyHC.
Minor points.
1. The over-expression and inhibition of miR-34b-5p have been added in Fig.1A, and the image quality has been improved. We hope this version will meet your requirement.
2. DM3 and DM5 means the 3rd and 5th days after myoblasts successfully induced to differentiate. MyoD (myogenic differentiation), MyoG (myogenin) and MyHC (myosin heavy chain) are three important marker genes during myogenesis, and the explanation has been added into line 160-161 and line163-165. In all figures, the image quality has been improved in the revised version.
3. Previous study have shown that miR-34b-5p could promote the proliferation of ALV-J infected DF-1 cells, and we didn’t investigate the proliferation role of miR-34b-5p in DF-1 cells. We only used DF-1 cell lines to do the Dual-Luciferase Reporter Assay and just want to prove the target relationship between miR-34b-5p and IGFBP2, and the rest experiments were all carried on chicken primary myoblast cells to investigate the function of miRNA and target genes on muscle development.
4. The mRNA and protein level of IGFBP2 overexpression have been added as Fig.4B,C, and the image quality of Fig4 has been improved.
5. The image quality of Fig5 has been improved and modified, and we hope that this version will meet your requirements.
6. This mistake and Figure has been corrected to accurately describe our results.
Reviewer 3 Report
The authors describe the role of miR-34 in regulating proliferation and differentiation of myoblast via IGFBP2. The role of microRNAs in regulating myoblasts proliferation and differentiation are more and more understood, however microRNA targets are yet to be explored for many microRNAs. miR-34 is a ubiquitously expressed microRNA with crucial roles in multiple tissues and in this manuscript, the authors present miR-34 role in muscle.
Overall, the manuscript is well written , however some data seem to be missing:
- Introduction is very short and does not summarise what has been achieved in the field so far, nor does it cite key references, eg. miR-34 role in muscle: MiR-34c represses muscle development by forming a regulatory loop with Notch1Scientific Reports volume 7, Article number: 9346 (2017) among others
- Changes in miR-34 expression following transfections are not presented – do the authors have this data?
- The labelling in the figures is too small
- In Fig1C the NC for mimic and inhibitor look very different, since these are controls, they should be roughly the same – can the authors comments on this?
- In Fig2B, miR-34 expression is upregulated at day 3 and Day 5 but not at Day 4 and Day 6 – can the authors comment on this, please?
- Fig2G only shows MyHC levels in control and mimic-treated cells, what about inhibitor-treated cells?
- In Fig3E – what is blank? Is it the mimic or inhibitor NC? Why have not both controls NC used like in other experiments?
- In Fig.4, the authors show western blotting for IGFBP2 after siIGFBP2 treatment but there is no western blot data showing overexpression of IGFBP2
- The authors investigated the levels of myogenin expression following miR-34 expression manipulations, however myogenin expression was not analysed after IGFBP2 level manipulation – can the authors comment on this?
- The schematic summary looks as if MyoD and MyHC repress myoblast differentiation – as this is not correct, perhaps this figure could be revised?
Author Response
Major points.
1. Sorry about our short introduction. Following your suggestions, we have added some new information in line 46-48 as follow: ‘It is also reported that miR-34a and miR-34c could inhibit the proliferation of smooth muscle cells [3, 4], and also miR-34c could repress the proliferation of muscle satellite cells while promote the differentiation process [5]’.
Ref:
[3] Chen Q.; Yang F.; Guo M.; Wen G.; Zhang C.; Le Anh L.; Zhu J.; Xiao Q.Zhang L. miRNA-34a reduces neointima formation through inhibiting smooth muscle cell proliferation and migration. J MOL CELL CARDIOL. 2015, 89,75-86. doi:10.1016/j.yjmcc.2015.10.017
[4] Choe N.; Kwon J.; Kim YS.; Eom GH.; Ahn YK.; Baik YH.; Park H.Kook H. The microRNA miR-34c inhibits vascular smooth muscle cell proliferation and neointimal hyperplasia by targeting stem cell factor. CELL SIGNAL. 2015, 27,1056-1065. doi:10.1016/j.cellsig.2014.12.022
[5] Hou L.; Xu J.; Li H.; Ou J.; Jiao Y.; Hu C.Wang C. MiR-34c represses muscle development by forming a regulatory loop with Notch1. SCI REP-UK. 2017, 7. doi:10.1038/s41598-017-09688-y
2. We have added the relative expression of miR-34b-5p after overexpression and inhibition in CPM cells in Fig.1A.
3. All of these images and figures have been improved and modified, and we hope this version will meet your requirement.
4. Mimic NC and inhibitor NC are used as negative control, theoretically, they have no effect on cells, visual differences caused by the images were because we want to show the influence of miR-34b-5p, so the picture we chosen may look like they have effects in myoblast proliferation and differentiation. Our statistics also showed that mimic NC and inhibitor NC have no significant difference between each other. Also, when we adjust the image quality we have changed and chosen another image and replaced the original EdU staining image of mimic NC to make it identify easily and roughly no difference in Fig.1B.
5. In this study, miR-34b-5p showed a relatively high expression in DM3 and DM5 but low in DM4 and DM6, probably because that miR-34b-5p not only could play an important role in myoblast differentiation, but also could be regulated by other regulatory factors to ensure that the differentiation process can proceed normally. Meanwhile, more evidence and investigation are required to explain this strange expression pattern.
6. We have added the western blot result of MyHC after miR-34b-5p inhibition as Fig.2H.
7. Blank means blank control, mimic NC and inhibitor NC are oligonucleotides designed by RiboBio (Guangzhou, China) and the sequence are secret, so here we used blank control to rule out the possibly that mimic or inhibitor NC could influence the IGFBP2 protein level. And our western blot result here proved that miR-34b-5p could directly regulate the expression of IGFBP2.
8. The mRNA and protein level of IGFBP2 overexpression have been added in Fig.4B,C.
9. We have also detected the MyoG expression level after IGFBP2 level manipulation and found no change, this maybe because the way miR-34b-5p regulate the myoblast differentiation are not totally the same with IGFBP2.
10 This mistake and Figure has been revised to more accurately describe our results.
Round 2
Reviewer 1 Report
The revised manuscript has been improved.
Unfortunately, the reproducibilityof the resluts is not clearly shown due to the limited time for revision.
Reviewer 2 Report
What I asked was done.